# Clinical Prediction Models for Recurrence in Patients with Resectable Grade 1 and 2 Sporadic Non-Functional Pancreatic Neuroendocrine Tumors: A Systematic Review

**DOI:** 10.3390/cancers15051525

**Published:** 2023-02-28

**Authors:** Jeffrey W. Chen, Charlotte M. Heidsma, Anton F. Engelsman, Ertunç Kabaktepe, Susan van Dieren, Massimo Falconi, Marc G. Besselink, Els J. M. Nieveen van Dijkum

**Affiliations:** 1Department of Surgery, Amsterdam UMC, Location University of Amsterdam, 1081 HV Amsterdam, The Netherlands; 2Amsterdam Center for Endocrine and Neuroendocrine Tumors (ACcENT), 1081 HV Amsterdam, The Netherlands; 3Cancer Center Amsterdam, 1081 HV Amsterdam, The Netherlands; 4Pancreatic Surgery, IRCCS Ospedale San Raffaele, Università Vita-Salute, 20132 Milan, Italy

**Keywords:** pancreatic neuroendocrine tumor, non-functional, resectable, recurrence, prediction model, grade 1, grade 2

## Abstract

**Simple Summary:**

The risk prediction for tumor recurrence after surgery for non-functional pancreatic neuroendocrine tumors is a major unmet clinical need. Accurate recurrence risk prediction could pave the way for tailor-made follow-up protocols in the future, as well as help select suitable patients for cancer treatment trials. Multiple prediction models have been developed; however, none are currently incorporated into international guidelines. This systematic review found 13 original models, of which 3 were validated outside the patient group in which they were developed. The effectiveness of a prediction model is not proven in the wider population without this validation; thus, the lack of it hinders the progress toward clinical use. We propose to test the included models in a large, multinational database to compare their performance. We recommend all authors developing a prediction model to perform the minimally required tests of model development and to implement it by creating an online calculator.

**Abstract:**

Recurrence after resection in patients with non-functional pancreatic neuroendocrine tumors (NF-pNET) has a considerable impact on overall survival. Accurate risk stratification will tailor optimal follow-up strategies. This systematic review assessed available prediction models, including their quality. This systematic review followed PRISMA and CHARMS guidelines. PubMed, Embase, and the Cochrane Library were searched up to December 2022 for studies that developed, updated, or validated prediction models for recurrence in resectable grade 1 or 2 NF-pNET. Studies were critically appraised. After screening 1883 studies, 14 studies with 3583 patients were included: 13 original prediction models and 1 prediction model validation. Four models were developed for preoperative and nine for postoperative use. Six models were presented as scoring systems, five as nomograms, and two as staging systems. The *c* statistic ranged from 0.67 to 0.94. The most frequently included predictors were tumor grade, tumor size, and lymph node positivity. Critical appraisal deemed all development studies as having a high risk of bias and the validation study as having a low risk of bias. This systematic review identified 13 prediction models for recurrence in resectable NF-pNET with external validations for 3 of them. External validation of prediction models improves their reliability and stimulates use in daily practice.

## 1. Introduction

Survival of patients with pancreatic neuroendocrine tumors (pNET) after resection is largely dependent on the development of tumor recurrence [1]. In contrast to other pancreatic neoplasms, pNET displays heterogeneous behavior with survival depending on tumor size, tumor grade, and treatment outcome. Patients with small (<2 cm) nonmetastatic grade 1 pNET have 5-year survival outcomes approaching 100%. Meanwhile, patients with larger metastasized grade 2 pNET had a 5-year survival rate of less than 40% [2,3]. Most patients are diagnosed incidentally with a non-functional pNET (NF-pNET), and only a subset of these patients eventually develops recurrence after tumor resection [4]. Surgical resection is the only curative intent treatment for localized pNET [5,6]. Improvements in the therapeutic armamentarium for advanced-stage disease in pNET have broadened the treatment options for patients with recurrence after surgery. Systemic therapies such as somatostatin analogs and chemotherapy appear to provide favorable outcomes in patients with metastases, while PRRT can be offered to a select group of patients [6,7,8,9].

Identifying patients at high risk for recurrence after resection with curative intent is a challenge faced by clinicians treating patients with a pNET. Currently, no tool is used to provide patients with standardized postoperative follow-up regimens [10]. Follow-up programs could be tailored according to the risk of recurrence, reducing the follow-up intensity of low-risk patients while those at high risk could undergo intensive surveillance. Furthermore, adjuvant therapy could also be tested in high-risk patients to prevent disease spread, but no such therapy is offered yet. Various studies have explored the risk factors associated with recurrence in patients with pNET [11,12,13]. Combining them into a prediction model is needed to make these tailor-made treatments possible. Multiple prediction models have been developed, but none of these models have been incorporated either into European Neuroendocrine Tumor Society (ENETS) guidelines or into clinical practice yet [7,14,15].

Accurate patient selection for the intensity of follow-up and (neo)adjuvant treatment is both an unmet need according to the ENETS guidelines, therefore warranting the need for a clinical prediction model [16]. The aim of this systematic review was to identify and evaluate currently available prediction models for recurrence in resected grade 1 and 2 NF-pNET and to select models for future use in clinical practice and clinical trials.

## 2. Materials and Methods

The systematic review followed the Critical Appraisal and Data Extraction for Systematic Reviews of Prediction Modelling Studies (CHARMS) guideline [17]. The study was reported according to the Preferred Reporting Items for Systematic Reviews and Meta-Analyses (PRISMA) guidelines [18]. The study was registered at PROSPERO (CRD42022380671).

### 2.1. Search Strategy

A systematic search was performed in MEDLINE via PubMed, Embase, and the Cochrane Library for studies published from inception until December 2022. The search included synonyms for [pancreatic neuroendocrine tumors] combined with [prognostic/predictive/prediction models] and [recurrence] (Appendix B). After the removal of duplicates, titles/abstracts and full-text articles were screened independently by two authors (J.W.C. and C.M.H.). Differences in opinion were resolved through discussion; if necessary, a third author (E.J.M.N.v.D.) was consulted. References of included articles were checked for other potentially eligible studies.

### 2.2. Eligibility

Studies developing, updating, or validating a clinical prediction model for recurrence in patients with grade 1 or 2 NF-pNET undergoing resection were included. Models designed for use in either the preoperative or postoperative setting were included. Studies providing an absolute probability estimate and those stratifying patients into risk categories were included, as both strategies might serve the clinical aim of providing more tailored treatment. To allow evaluation and comparison of model performance, at least one of the following performance outcomes had to be reported: *c* statistic, area under the curve (AUC), R2, Brier score, sensitivity, specificity, calibration plots, or calibration statistics.

Review papers, abstracts, case studies, studies with patients aged less than 18 years, and non-human studies were excluded. Articles including patients with diseases other than pancreatic neuroendocrine tumors (such as pancreatic ductal adenocarcinoma or intraductal papillary mucinous neoplasm) or other neuroendocrine tumor localizations were excluded. Studies were excluded if more than 20% of its population were patients with WHO grade 3, genetic background, metastatic or functional pNET. If not reported, the studies were included but marked as a risk of bias. Studies validating a staging system (i.e., TNM, WHO, AJCC) without any modification were not included, because the limited discriminative strength of these systems was one of the reasons for developing better models [19,20,21].

### 2.3. Data Extraction and Analysis

Data extraction and critical appraisal were performed independently by two authors (J.W.C. and C.M.H.). Articles were categorized into development/model update studies and validation studies. A data extraction sheet was used to extract data for all studies and included: the first author, year of publication, pre- or postoperative setting, sample size, study interval, source of data, countries of inclusion, number of centers, predicted outcome, included predictors, model performance, information on validation and number of citations. For development/update studies, model development method, number of prognostic factors screened, and final model presentation were additionally extracted. Predictors included in regression analyses were collected and scored for statistical significance. If multiple models were developed or tested, the prediction model with the highest *c* statistic or the model proposed by the authors was included. Meta-analysis was not possible due to heterogeneity in prediction model variables and outcomes. Critical appraisal was performed following the CHARMS guidelines [22]. The PROBAST critical appraisal tool was used for the assessment of methodological quality [23]. A clinical epidemiologist was consulted for this review [S.v.D.].

### 2.4. Definitions and Terminology

A prediction model was defined as “a formal combination of multiple predictive factors from which risks of a specific endpoint can be calculated for individual patients” [24]. Clinical prediction models should “discriminate between individual patients who do and do not experience a specific event (discrimination), should make accurate predictions (calibration), perform well across different patient populations (generalizability) and be readily interpretable” [25,26].

Model performance can be assessed by its ability to discriminate and calibrate. Discrimination (i.e., do patients who have the outcome also have a higher predicted risk than those who do not have the outcome) can be quantified through measures such as sensitivity, specificity, area under the receiver operating curve, or by the concordance statistic (*c* statistic) [27,28]. A *c* statistic of 1.0 represents a perfect model, while a score of 0.5 indicates that the model is not better at prediction than random selection [29]. In binary outcomes, the AUC is equal to the *c* statistic. In turn, calibration is important for model performance since it compares the predicted probability of the outcome (i.e., recurrence) with the actual outcome. It is most often visualized using calibration plots or assessed as goodness of fit, which can be quantified using the Hosmer–Lemeshow test (*p* < 0.050 indicates poor calibration).

Model validation is imperative and can be performed using several different techniques. It is possible to internally validate (reproducibility) followed by external validation (generalizability). Internal validation can be performed through split-sample validation (training and test set), cross-validation (development in random segments of the population, tested in the remaining segment and repeating this process), and bootstrapping (random samples of the same size are drawn with replacement) [24,29]. Ideally, all prediction models should undergo not only internal but also external validation. Geographical (using a new cohort from a different center), temporal (same center, but patients at a different time interval), and fully independent (new research group at a different center) are the three most important external validation options [25,29].

The term tumor grade was used for both the World Health Organization (WHO) definition of tumor grade as well as the Ki-67 index, unless specified otherwise. Tumor grade was defined according to the WHO 2017 classification: grade 1 (Ki-67 index of <3%), grade 2 (Ki-67 index of 3–20%), and grade 3 (Ki-67 index of >20%), unless otherwise specified [20]. Histological grade was defined as the grading of tumor cell differentiation into well, intermediate, or poorly differentiated.

## 3. Results

### 3.1. Baseline Characteristics

The search yielded 1883 studies (Figure 1), of which 14 could be included in this review containing 3583 patients, of which 1668 (46.6%) were female. The median age was 58 years (IQR 53–60). The median follow-up duration was 48 months (IQR 40–56). Model development data from 3241 patients were available for this systematic review. Models were developed with a median number of 140 patients (IQR 87–235) consisting of 83.0% non-functioning pNET cases, 6.7% metastatic, and 7.9% genetic. The population consisted of 1527 patients (62.7%) with a WHO grade 1 tumor, 836 patients (34.3%) with a grade 2 tumor, and a small percentage of patients had grade 3 (*n* = 73, 3.0%). No patients received (neo)adjuvant therapy. The most commonly performed procedure was distal pancreatectomy (*n* = 1179, 46.7%), followed by pancreatoduodenectomy (*n* = 696, 27.6%), enucleation (*n* = 233, 9.2%), central pancreatectomy (*n* = 95, 3.8%), and total pancreatectomy (*n* = 53, 2.1%). The 5-year RFS was reported by seven studies, reporting a median survival of 74% (IQR 72.0–84.6) [30,31]. The 5-year overall survival was reported by six studies with a median survival of 90.6% (IQR 80.1–91.0) [32,33,34,35,36,37]. Full details of other characteristics can be found in Table 1.

### 3.2. Preoperative Prediction Models

Four out of thirteen prediction models were designed for preoperative use [31,33,36,37], with *c* statistics ranging from 0.78 to 0.94. Sun et al. used preoperative MRI-imaging variables to calculate the recurrence risk [31]. Tumors exhibiting hypoenhancement and low apparent diffusion coefficient values were associated with worse RFS after curative resection. Fisher et al. included serum chromogranin A > 5 times the upper limit and the presence of a recurrent tumor as recurrence predictors [33]. Their sensitivity analysis suggested that chromogranin A might have a low predictive value in low-grade tumors. Primavesi et al. included C-reactive protein ≥ 0.2 mg/dL and the presence of distant metastasis as predictors [36]. However, in their study, C-reactive protein was not associated with inferior 5-year RFS (72.2% vs. 70.4%, *p* = 0.89) or increased recurrence rate (25.6% vs. 21.2%, *p* = 0.34). The model by Zhou et al. was designed for preoperative prediction of both recurrence and survival, but only the AUC of overall survival was reported (0.83) [37]. Their study analyzed the predictive strength of “gamma-glutamyl transferase/lymph-node-ratio” as a preoperative predictor, but the predictive strength of adding this predictor to the AJCC staging system (AUC 0.81 [95% CI 0.73–0.90]) or WHO classification system (AUC 0.81 [0.73–0.88]) was inferior to the combination of the AJCC staging system and WHO classification (AUC 0.83 [0.75–0.91]). Full details on prediction model performance can be found in Table 2.

### 3.3. Postoperative Prediction Models

Nine prediction models were developed for postoperative use [11,30,32,34,35,38,39,40,41]. The *c* statistics ranged from 0.75 to 0.94. Four studies created a model based on a combination of tumor grade, tumor size, and lymph node positivity (Table 2) [11,30,35,40]. The most frequently included predictor was tumor grade (n = 11) [11,30,34,35,38,39,40,41]. All but two studies followed the WHO 2017 cut-off values for tumor grade [20]. Pulvirenti et al. incorporated the Ki-67 index as a continuous variable in their nomogram [11]. Sho et al. used a Ki-67 index <8% as a low tumor grade cut-off value for its scoring system [35]. Tumor size was the second most included predictor (n = 11) [11,30,32,35,40,41]. Two models incorporated tumor size as a continuous variable [11,40], two models used a cut-off value of ≥5 cm [32,35], one used a cut-off value of ≥2.5 cm [41], and one model incorporated tumor size into a tumor burden score [30]. Viúdez et al. developed the Immunohistochemistry Prognostic Score as a predictor (high Immunohistochemistry score; HR 2.68 [95% CI 1.60–4.49], *p* = 0.03), where they quantified the immunohistochemistry patterns of DNA-methylation [42]. Wei et al. also developed their own predictor, i.e., the Immunoscore (low Immunoscore; HR 0.13 [0.06–0.39], *p* < 0.001), where the immune response in the tumor microenvironment was quantified [39]. Furthermore, Ballian et al. and Liu et al. incorporated the Hochwald grading system as a predictor, where a mitotic rate of >2 per 50 high-power field and signs of histopathological necrosis were associated with a higher recurrence rate [32,38,42]. However, association through multivariable analyses was not reported in both studies.

### 3.4. Predictor Selection

Predictor selection based on uni- and multivariable Cox regression analysis was performed in nine studies (Table 1) [31,34,35,37,38,39,40,41,43]. Three studies selected predictors through univariable Cox regression analysis [11,32,33] and Dong et al. through logistic regression analysis [30]. Although stating that the model was built on the beta coefficients of multivariable analysis, only the results of their univariable analysis were reported by Dong et al. [30,44]. The included studies screened a median of 11 variables (range 9–17) and included a median of 3 variables (range 2–7) in their final prediction model. Tumor grade was most frequently associated with RFS in 10 out of 11 included analyses after regression analysis (91%) [11,30,33,34,35,37,38,39,40,41] (Table 3), followed by tumor size (73%) [11,31,32,33,35,38,40,41]. The patients’ sex was analyzed by seven studies, but only one study showed an association with RFS (Male; HR 2.22 [95% CI 1.14–4.31], *p* = 0.018) [37].

### 3.5. Grading

Heidsma et al., Pulvirenti et al., and Zou et al. all reported grading following the 2017 WHO guidelines [11,40,43]. Two studies failed to report which grading guideline was used [33,39]. Two used the Hochwald criteria for histologic grading [32,38]. The other studies followed the 2010 WHO tumor grading system.

### 3.6. Discrimination

Discrimination of the prediction models, expressed in *c* statistic or AUC, was described in all studies (Table 2). However, Zhou et al. only provided the *c* statistic for OS, despite presenting their model as a predictor for recurrence also [37]. The *c* statistic for predicting RFS ranged between 0.67 and 0.94. The *c* statistic for predicting OS ranged from 0.69 to 0.83 [36,37,41]. Primavesi et al. tested their model development for three different outcomes (RFS, DSS, OS), of which it worked best for the prediction of DSS (77.3 [67.2–87.5]). However, it had the lowest discrimination score of the included models, with an AUC of 66.5 [36]. The highest discrimination was reported by the prediction model of Zou et al., with a *c* statistic of 0.94 for 5-year RFS [40].

### 3.7. Calibration

Calibration was reported by seven studies [11,30,31,34,39,40,43]. Although all studies concluded that their model had good agreement of calibration, differences were present. The calibration curves presented by Pulvirenti et al. and Wei et al. showed a relatively high tendency to underpredict the risk of recurrence compared to the other models [11,39]. Genç et al. did not present their calibration curves but did provide a Hosmer–Lemeshow test with a Chi-square of 11.25, *p* = 0.258, indicating good calibration [34].

Four studies performed an internal validation using bootstrapping [11,30,35,39]. One performed a split-sample validation and had a median *c* statistic of 0.78 (range 0.74–0.84) [33]. Two studies reported the *c* statistic both before and after internal validation [30,41]. Dong et al. performed a bootstrapping method with 5000 iterations and went from a *c* statistic for RFS of 0.75 (95% CI 0.66–0.79) to 0.71 (0.65–0.75). Viúdez et al. reported a *c* statistic of 0.80 before and 0.78 after internal validation. The *c* statistic for OS was 0.79 before and 0.76 after internal validation.

Three out of thirteen original prediction models were externally validated [11,34,39]. The prediction model by Genç et al. was externally validated by Heidsma et al. [34,43]. The *c* statistic of the development model was 0.81 (95% CI 0.75–0.87) and 0.77 (0.71–0.83) in the validation model. Heidsma et al. performed a sub-analysis in patients with a tumor > 2 cm, in which the model performed better with a *c* statistic of 0.79 (0.73–0.85). Pulvirenti et al. developed a model with a *c* statistic of 0.85, which remained comparable after externally validating it in a separate international cohort (*c* statistic 0.84 [0.79–0.88]) [11]. The development model by Wei et al. had a *c* statistic of 0.92 (0.88–0.95) and a *c* statistic of 0.86 (0.80–0.93) in the external validation [39].

### 3.8. Critical Appraisal

Methodological assessment of the included studies showed poor overall quality, primarily in the Analysis domain (Appendix A). All developed models were scored positive for risk of bias. Most studies did not have the minimally required number of patients with an event (recurrence) in the development cohorts for the prevention of overfitting. Dong et al. and Fisher et al. were the only two models with enough events per variable, 66 events and 60 events, respectively [30,33]. However, Fisher et al. developed their preoperative prediction model using postoperatively determined tumor grade [33]. Only the models reported by Genç et al., Pulvirenti et al., and Wei et al. were externally validated [11,34,39]. The models developed by Wei et al. and Viúdez et al. were flagged for concerns of applicability due to the need for highly specific predictors (Immunoscore and Immunohistochemistry Prognostic Score, respectively) in order to use these models in daily clinical practice [39,41].

## 4. Discussion

This first systematic review to evaluate existing prediction models for recurrence in patients with resectable grade 1 and 2 NF-pNET found 13 model development studies and one validation study. Most models were presented as scoring systems, followed by nomograms and modified staging systems. The most frequently incorporated risk factor was tumor grade, which had the highest rate of significant association with recurrence after regression analyses (91%). However, 10/13 (76.9%) models were not externally validated, thereby hindering the progress towards clinical implementation. The studies that did perform an external validation also performed the minimally required performance tests of prediction model development, i.e., discrimination, calibration, internal validation, and external validation.

Taking the outcomes of the critical appraisal into account, the results of the development models by Genç et al. (*c* statistic 0.77), Wei et al. (0.86), and Pulvirenti et al. (0.84) appear to be the most reliable since these are the only studies that performed the minimally required tests [11,34,39]. The predictors used by these three models were tumor grade/Ki-67 index, positive lymph nodes, tumor size, vascular and/or perineural invasion, metastasis, and ‘Immunoscore’. Tumor grade was incorporated in all three models. The incorporated predictors are in line with the significant predictors described in the literature [45].

Genç et al. and Wei et al. both dichotomized their continuous variables, including the Ki-67 index [34,39]. Dichotomizing a continuous variable leads to a loss of predictive ability due to an assumption of a constant level of risk above and below the threshold [46]. A separate study by Genç et al. showed that variations within the margins of WHO grade 2 (Ki-67 index 3–20%) led to significant differences in recurrence rate [12]. Lopez-Aguiar et al. reported that these significant changes in prognosis even occur within the margins of grade 1 (0–2.99%) [47]. In turn, dichotomized variables are user-friendlier than continuous variables and will likely result in the more frequent use of the prediction model. For instance, frequent internationally used models, such as the CHA2DS2VASc-model for calculating stroke risk for patients with atrial fibrillation [48] and the Wells’ criteria model for predicting the risk of a pulmonary embolism [49], also use dichotomized variables. As such, the loss of the predictive ability of the variable does not outweigh the benefit of dichotomizing it since one of the major pitfalls of prediction models is that they are seldom used in clinical practice.

Wei et al. developed a unique predictor where they quantified the immune response in the tumor microenvironment into the Immunoscore [39]. A low Immunoscore, i.e., a pattern of low peritumoral inflammatory activity and high intratumoral CD8+ activity, was associated with a better RFS. Several studies have reported significant associations between immune response patterns in the tumor microenvironment and NET prognosis [50,51,52]. Takahashi et al. found that certain immune patterns were complementary to the WHO 2017 grading system, providing an argument for a possible augmenting effect if combined [53]. Immunoscores could be a powerful addition in the future; however, the scarcity of literary evidence speaks against suggesting this model for immediate implementation. Moreover, the highly specific tests required to determine the immune response patterns and the risk of interobserver variations make it less likely that the model would be internationally used in daily clinical practice.

Contrary to the other models, Pulvirenti et al. preserved the continuous function of its numeric predictors in their nomogram, i.e., Ki-67 index, tumor size, and positive lymph nodes [11]. A possible confounding factor is that they combined vascular invasion and perineural invasion into a single predictor, while the variables were associated with different hazard ratios (8.55 [95% CI 5.14–14.21] and 5.91 [3.72–9.40], respectively). Thereby risking underprediction of RFS when only vascular invasion is present, overprediction when only perineural invasion is present, and losing predictive strength when both predictors are present. Additionally, calibration of the nomogram by Pulvirenti et al. showed a tendency to underpredict if the 5-year RFS was between 55 and 80%. However, in clinical practice, their model could be used to identify patients with low-risk tumors (>80% chance of 5-year RFS) that would benefit from a low-frequency postoperative follow-up.

Selecting the correct population is particularly important in models predicting outcomes in pNET. They exhibit heterogeneous behavior, which challenges accurate risk stratification [54]. Current guidelines now recommend watchful waiting for asymptomatic NF-pNET ≤2 cm due to its favorable prognosis [7,14,55]. Yet, a total of 39.8% of patients from 8 of the development cohorts in this review had a tumor of ≤2 cm [31,34,35,37,38,39,40,41]. The model by Genç et al. was additionally externally validated by Heidsma et al. for pNET >2 cm and resulted in better discrimination [34,43]. The risk of recurrence is likely to be underestimated by a prediction model if a large proportion of indolent tumors are included. Future studies should exclude these small, low-grade tumors to increase the relevance of its population when studying resectable cases or perform a separate analysis for this group.

The main practical advantage of using a prediction model is its ability to discriminate patients with a low risk for recurrence from high-risk patients. The recurrence risk threshold up to which pNET can be considered low-risk tumors differed among the studies included in this review from 3.1% to 19.9% [11,31,33,35,37,38,39,40,43]. Prospective studies using recurrence prediction models are needed to determine the optimal cut-off value, as well as the optimal timing for postoperative imaging.

The results of this review should be interpreted considering several limitations. First, the studies included in this systematic review are prone to publication bias since poorly performing prediction models are rarely publicized. Second, a frequent risk of bias was the low number of events per variable due to small sample sizes. For the screening of a single predictor, the rule of thumb states that at least 10 events are required to eliminate bias and preferably more [56,57,58]. However, the studies developing a prediction model for recurrence screened a median of 11 predictors, while the median number allowed was less than 2. This means that the predictors that were selected are susceptible to overfitting. Third, predictor selection based on univariable analysis could result in omitting potentially relevant predictors or inclusion on the basis of accidental association. For example, as seen in Table 3, tumor grade shows a strong association with RFS. The omission of this predictor by Sun et al., due to it not reaching statistical significance after univariable analysis, probably resulted in a less effective prediction model [31]. Preferably, predictors should be chosen based on their relationship to the predicted outcome and not solely on the basis of their statistical significance [59]. However, the use of univariable analysis to choose predictors for the multivariable analysis is common practice and, if handled correctly, should not pose a problem for the reliability of a model. Fourth, the most important bias-introducing factor was the lack of external validation. The stated accuracy of a prediction model cannot be made certain without it, thus limiting the applicability of the results of this systematic review. It provides a strong argument for multicenter collaboration when developing and validating prediction models for patients with an NF-pNET. It is otherwise nearly impossible to obtain sufficient events (i.e., recurrence) to create a reliable prediction model.

The main strength of the current systematic review is the strict list of eligibility criteria that was used to create a homogeneous study population, thereby limiting confounding factors that influence the risk of recurrence. This study also provides an overview of the current landscape of developed prediction models as well as an overview of the predictive strength of the analyzed and included predictors. This information could be used for the selection of predictors for multivariable regression analysis when developing a prediction model.

## 5. Conclusions and Future Directions

Despite the development of several prediction models for the recurrence of NF-pNET after surgery, they are rarely evaluated on their performance in other participant data. To judge the true performance of a prediction model, it must be externally validated to evaluate model overfitting or deficiencies in the statistical modeling [29,60]. None of the models have undergone impact analysis in prospective studies. This might be a reason why none of the models are incorporated into (international) guidelines. Preferably, a model should be investigated in multiple external validation studies and later, ideally, also in a (randomized) controlled trial [61]. We propose to test the included prediction models in a large, multinational cohort to compare their performance. The best-performing prediction model could thereafter be applied in both clinical and trial settings to determine risk outcomes in pNET. In turn, models presented as an online calculator may be the most suitable for regular use since they are easier to access for clinicians and insightful for patients. The only model currently available online is the model by Genç et al. [34]. We highly recommend all authors developing a prediction model to calibrate, discriminate, and validate their results and to implement the model by creating an online calculator.

## Figures and Tables

**Figure 1 cancers-15-01525-f001:**
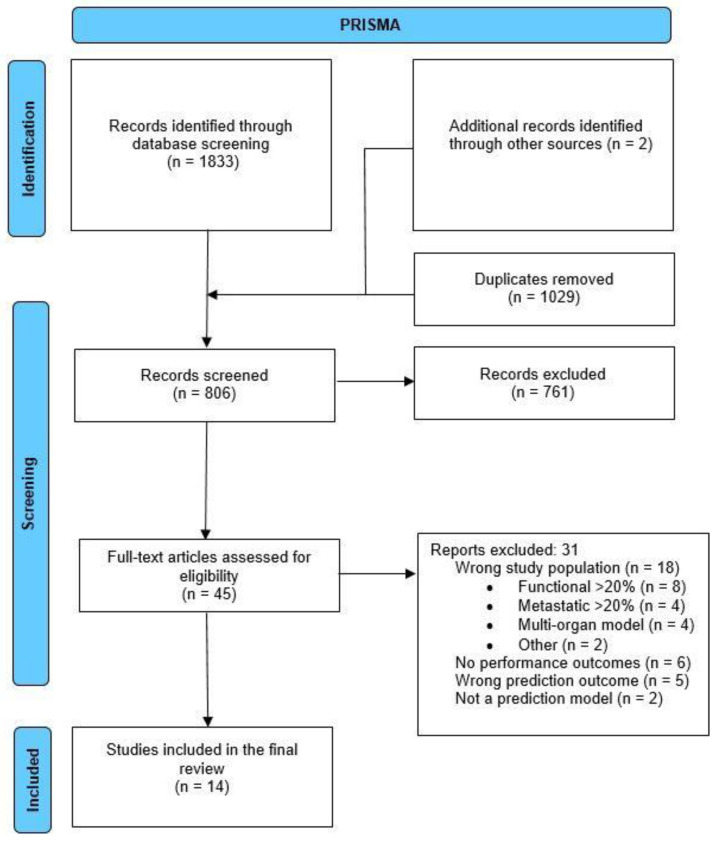
Preferred Reporting Items for Systematic Reviews and Meta-Analysis (PRISMA) flow diagram reporting the number of studies identified, screened, and selected for inclusion in this systematic review.

**Table 1 cancers-15-01525-t001:** Prediction model characteristics.

Author (Year)	n	Setting	Country	Single or Multicenter	Study Interval	Model Development Method	Study Type	Follow-Up (IQR)	Outcome Predicted	Definition of Outcome
Ballian et al. (2009)	43	Postop	USA	Single center	1991–2007	Univariable Cox regression analysis	DEV	68 months	RFS (5-year) DSS (5-year)	Radiographic evidence of new tumors
Dong et al. (2021)	416	Postop	China	8 centers	1997–2016	Uni- and multivariable logistic regression analysis	DEV	31 months (11.3–56.6)	RFS (5-, 10-year)	Time from surgery to the time of identification of suspicious imaging findings or biopsy-proven tumor
Fisher et al. (2019)	224	Preop	USA	8 centers	2000–2016	Univariable Cox regression analysis	DEV	37 months (15–62)	RFS	Time of resection until the time of radiographic or pathological evidence of tumor recurrence
Genç et al. (2018)	211	Postop	Netherlands, Italy	3 centers	1992–2015	Uni- and multivariable Cox regression analysis	DEV	51 months (29–72)	RFS (5-year) DSS (5-year)	RFS: Percentage of patients without recurrence in the pancreas, new positive lymph nodes, or distant metastasis after resection DSS: Percentage of patients who have not died due to pNET
Heidsma et al. (2021)	342	Postop	Australia, Austria, France, Germany, Netherlands, Sweden, USA	7 centers	1991–2018	Uni- and multivariable Cox regression analysis	VAL	50.5 months (22.3–103)	RFS (5-year)	Date of the first cross-sectional imaging on which a new local or metastatic lesion was detected
Liu et al. (2013)	75	Postop	USA	Single center	1993–2009	Uni- and multivariable Cox regression analysis	DEV	69 months (range 1–212)	RFS	Time from surgery to death due to disease or to disease recurrence at local, regional, or distant sites, whichever occurred first
Primavesi et al. (2020)	D: 160 V: 204	Preop	Austria, Germany, Italy, Netherlands	D: 6 centers V: 4 centers	D: 1998–2017 V: 1990–2018	Uni- and multivariable Cox regression analysis	DEV	57.5 months (28.3–83.3)	RFS (5-, 10-year) DSS (5-, 10-year) OS (5-, 10-year)	RFS: Time from initial curative intent to recurrence/last follow-up DSS: Time to pNET related-death/last follow-up OS: Time from first surgery to death/last follow-up
Pulvirenti et al. (2021)	D: 632 V: 328	Postop	Australia, UK, USA		2000–2016	Univariable Cox regression analysis	DEV	51 months	RFS (5-year)	Date of curative surgery until date of first recurrence identified through routine postoperative CT-scans
Sho et al. (2019)	140	Postop	USA	Single center	1989–2015	Uni- and multivariable Cox regression analysis	DEV	56 months	RFS (5-year)	Time to the last known date the patient was disease free
Sun et al. (2019)	81	Preop	China	Single center	2009–2017	Uni- and multivariable Cox regression analysis	DEV	16 months (range 6–108)	RFS (1-, 2-, 3-year)	Day of surgery to the time of local recurrence or distant metastatic disease on radiological images, last clinical follow-up, or death
Viúdez et al. (2016)	92	Postop	USA	Single center	1998–2010	Uni- and multivariable Cox regression analysis	DEV	n.r.	RFS OS	Time of surgery to date of relapse, death, or last follow-up
Wei et al. (2021)	D: 125 V: 77	Postop	China	Single center	2012–2018	Uni- and multivariable Cox regression analysis	DEV	41 months (27–59.8)	RFS (3-, 5-year)	n.r.
Zhou et al. (2017)	125	Preop	China	Single center	2003–2016	Uni- and multivariable Cox regression analysis	DEV	45.8 months (SD 37.01)	RFS OS	RFS: Time from date of surgery to date of recurrence OS: Time from date of initial diagnosis until date of death from any cause or date of last known contact
Zou et al. (2020)	245	Postop	China	Single center	2002–2018	Uni- and multivariable Cox regression analysis	DEV	40 months	RFS (3-, 5-year)	RFS: Date of recurrence in any forms

IQR, interquartile range; Postop, postoperative; Preop, preoperative; USA, United States of America; UK, United Kingdom; D, development cohort; V, validation cohort; DEV, development study; VAL, validation study; OS, overall survival; DSS, disease-specific survival; RFS, recurrence-free survival; SD, standard deviation; n.r., not reported.

**Table 2 cancers-15-01525-t002:** Prediction model performance.

Author (Year)	Model Type	No. of Variables Screened	Predictors in Final Model	Discrimination, *c* Statistic (95% CI)	Calibration	Internal Validation	External Validation
Ballian et al. (2009)	Scoring system	9	Tumor size ≥ 5 cm, Hochwald grading system *(mitotic index + necrosis),* positive lymph node, R1 resection	RFS: 0.80 (0.68–0.91) DSS: 0.81 (0.73–0.90)	n.r.	n.r.	n.r.
Dong et al. (2021)	Nomogram	10	Tumor grade *(2010),* tumor burden score *((max. tumor diameter)*^2^* + (number of tumors)*^2^*)*, positive lymph node	0.75 (0.66–0.79)	Good performance	Bootstrapping: 5000 iterations 0.71 (0.65–0.75)	n.r.
Fisher et al. (2019)	Scoring system	6	Tumor grade 2 or 3, chromogranin A > 5× upper limit, surgery for tumor recurrence, tumor size ≥ 4 cm	See internal validation	n.r.	Split-sample validation AUC: 0.78	n.r.
Genç et al. (2018)	Nomogram	11	Tumor grade *(2010)*, positive lymph node, perineural invasion	0.81 (0.75–0.87)	GOF; Hosmer Lemeshow Chi-square 11.25, *p* = 0.258	Performed	n.r.
Heidsma et al. (2021)	Nomogram	N/A	Tumor grade *(2017)*, positive lymph node, perineural invasion	0.77 (0.71–0.83)	Calibration slope 0.74	N/A	N/A
Liu et al. (2013)	Staging system	9	Ki-67 index, Hochwald grading system *(mitotic index + necrosis)*	0.79	n.r.	n.r.	n.r.
Primavesi et al. (2020)	Scoring system	7	CRP > 0.2 mg/dL, tumor size > 3 cm, metastasis	RFS: AUC 66.5 (58.9–74.2) DSS: AUC 77.3 (67.2–87.5) OS: AUC 68.9 (61.5–76.4)	n.r.	n.r.	n.r.
Pulvirenti et al. (2021)	Nomogram	11	Number of positive lymph nodes, Ki-67 index, tumor size, vascular and/or perineural invasion	0.85	Performed	Bootstrapping: 100 iterations	0.84 (0.79–0.88)
Sho et al. (2019)	Scoring system	13	Tumor size ≥ 5 cm, positive lymph node, tumor grade *(2010)*	0.82 (0.72–0.92)	n.r.	Bootstrapping: 1000 iterations	n.r.
Sun et al. (2019)	Nomogram	22	Tumor size > 2 cm, hypoenhancement, apparent diffusion coefficient	0.91 (0.84–0.98)	Performed	n.r.	n.r.
Viúdez et al. (2016)	Scoring system	14	RFS: AJCC, tumor size, tumor grade, Immunohistochemistry Prognostic Score *(MGMT, PHLDA-3, NDRG-1 expressions)* OS: AJCC, age > 60, Immunohistochemistry Prognostic Score *(MGMT, PHLDA-3, NDRG-1 expressions)*	RFS: 0.80 OS: 0.79	n.r.	n.r.	n.r.
Wei et al. (2021)	Nomogram	20	Metastasis, tumor grade, Immunoscore *(0.261 × the status of CCL19) + (0.490 × the status of IL-16) + (0.123 × the status of CD163) + (0.044 × the status of CD8PT)–(0.011× the status of CD8IT)–(0.493× the status of IRF4)*	0.92 (0.88–0.95)	Good agreement	Bootstrapping	0.86 (0.80–0.93)
Zhou et al. (2017)	Staging system	27	AJCC, tumor grade *(2010)*	OS: AUC 0.83 (0.75–0.91)	n.r.	n.r.	Not performed
Zou et al. (2020)	Scoring system	11	Positive lymph node, tumor size, tumor grade *(2017)*	3 year: 0.91 5 year: 0.94 8 year: 0.93	Good performance	n.r.	n.r.

GOF, Goodness-of-Fit; N/A, not applicable; CRP, C-reactive protein; DSS, disease-specific survival; RFS, recurrence-free survival; OS, overall survival; n.r., not reported; AJCC, American Joint Committee on Cancer.

**Table 3 cancers-15-01525-t003:** Predictor association with recurrence-free survival.

	Tumor Grade/ Ki-67 Index	Tumor Size	Positive Lymph Node	Sex	Age	Perineural Invasion	R1 Resection	Vascular Invasion	Histological Grade	Functional Tumor	Symptomatic	Distant Metastasis	TNM-Stage	Minimally Invasive	Predictors Screened
Study															
Pulvirenti (2021) ^b^	**✓**	**✓**	**✓**	**X**	**✓**	**✓**	**✓**	**✓**		**✓**				**X**	10
Dong (2021) ^b^	**✓**		**✓**	**X**	**X**	**X**	**X**		**✓**		**X**			**✓**	9
Liu (2013) ^a^	**✓**	**✓**	**✓**	**X**	**X**		**X**	**✓**		**X**					8
Ballian (2009) ^b^		**✓**	**✓**	**X**	**X**	**✓**	**✓**	**X**							7
Zhou (2017) ^a^	**✓**	**X**		**✓**		**X**					**X**	**X**	**✓**		7
Primavesi (2020) ^a^		**X**		**X**	**✓**						**X**	**✓**			5
Zou (2020) ^a^	**✓**	**✓**		**X**		**X**		**X**							5
Fisher (2019) ^b^	**✓**	**✓**			**X**					**X**					4
Sho (2019) ^a^	**✓**	**✓**	**✓**						**✓**						4
Sun (2019) ^a^	**X**	**✓**	**X**						**X**						4
Genç (2018) ^a^	**✓**		**✓**			**✓**									3
Viúdez (2016) ^a^	**✓**	**✓**											**X**		3
Wei (2021) ^a^	**✓**	**X**	**X**												3
Predictor analyzed by study (out of 13), n (%)	11 (85)	11 (85)	8 (62)	7 (54)	6 (46)	6 (46)	4 (31)	4 (31)	3 (23)	3 (23)	3 (23)	2 (15)	2 (15)	2 (15)	
Predictor significance, n (%)	10/11 (91)	8/11 (72)	6/8 (75)	1/7 (14)	2/6 (33)	3/6 (50)	2/4 (50)	2/4 (50)	2/3 (67)	1/3 (33)	0/3 (0)	1/2 (50)	1/2 (50)	1/2 (50)	

^a^ Studies that performed both uni- and multivariable analysis; ^b^ Studies that performed only univariable analysis; **✓** = statistically significant association; **X** = no statistically significant association; AJCC American Joint Committee on Cancer.

## Data Availability

All data are provided within this article or Appendix A.

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
