# Peer review of "Clinical Prediction Models for Recurrence in Patients with Resectable Grade 1 and 2 Sporadic Non-Functional Pancreatic Neuroendocrine Tumors: A Systematic Review"

_cancers, 2023, doi:10.3390/cancers15051525_

Round 1

Reviewer 1 Report

This paper is a systematic review of clinical prediction models for recurrence in patients with resctable G1-2 NF-panNETs. It presents all data from available studies in a conprehensive and detailed way and critically assesses their applicability and risk of bias. The need of external validation of each proposed model as well as the sufficient power, in terms of participant numbers and events, is correctly highlighted. The conclusions of this review should be a guide for future similar studies testing prediction models in various site NETs, although the heterogeneity of this disease frequently hampers safe conclusions. A few minor suggestions/comments:

1. line 61: Furthermore, adjuvant therapy could also be tested ...

2. lines 104-105: "Studies consisting more than 20% of patients with WHO grade 3, genetic background, metastatic or functioning panNET were also excluded". Please clarify if >20% refers only to WHO g3 tumours or in all other criteria used (i.e. were studies with >20% of genetic pan-NETs excluded or the authors excluded any study recruiting any patients with genetic pan-NET. if so, maybe the word sporadic could be used in the article title). In the same sense some studies in table 3 had metastatic and functional tumours.

2. Please rephrase "in more use" (line 319)

3. I can conceive the importance of a post-operative predictor model for recurrence, but can the authors elaborate on the significance of pre-operative validation, since pathologoanatolomical data are lacking and the effect of the intervention (ie surgery) is not taken into account.

4. As the authors mention the main practical advantage that these prediction models offer is the discrimination of patients with low probabilty of recurrence for less frequent follow-up. In this case in my opinion the authors could comment on the probability threshold used to make such a discimination.

Reviewer 2 Report

Thank you for a very interesting and well-written manuscript. 

Only comment is to check the manuscript for typos etc and then resubmit

Reviewer 3 Report

The homogeneity and data integrity of the included studies were too low and there was considerable bias. It is recommended that the number of included studies be further expanded

Round 2

Reviewer 3 Report

It will be better if the authors can distinguish postoperative patients and make predictive models, it may be better able to guide clinical practice
